# MYNA: MASKING-BASED CONTRASTIVE LEARNING OF MUSICAL REPRESENTATIONS

## ABSTRACT

In this paper, we present Myna, a simple yet effective approach for self-supervised musical representation learning. Built on a contrastive learning framework, Myna introduces two key innovations: (1) the use of a Vision Transformer (ViT) on mel-spectrograms as the backbone, replacing SampleCNN on raw audio; and (2) a simple yet novel data augmentation strategy—token masking—that masks 90% of spectrogram tokens (e.g., 16x16 patches). These innovations deliver both effectiveness and efficiency: (i) Token masking enables a significant increase in per-GPU batch size, from 48 or 120 in traditional contrastive methods (e.g., CLMR, MULE) to 4096. (ii) By avoiding traditional augmentations (e.g., pitch shifts), Myna retains pitch sensitivity, enhancing performance in tasks like key detection. (iii) The use of vertical patches (128x2 instead of 16x16) allows the model to better capture critical features for key detection. Our hybrid model, Myna-22M-Hybrid, processes both 16x16 and 128x2 patches, achieving state-of-the-art results. Trained on a single GPU, it outperforms MULE (62M) on average and rivals MERT-95M, which was trained on 16 and 64 GPUs, respectively. When scaled to 85M parameters, Myna achieves further improvements across all tasks and is competitive with models like MERT-330M, MusicFM, and MuQ despite being 3-7x smaller and trained with an order of magnitude fewer GPUs in less time. Additionally, it surpasses MERT-95M-public and $MuQ_{m4a}$, establishing itself as the best-performing model trained on publicly available data. We release our code and models to promote reproducibility and facilitate future research: https://github.com/ghost-signal/myna

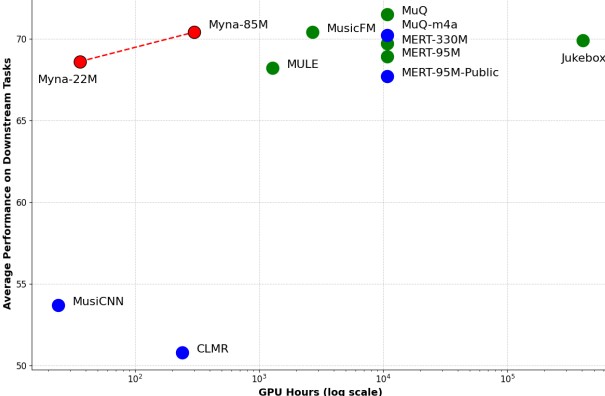

Figure 1: Myna is efficient: we achieve competitive downstream task performance while requiring significantly fewer computational resources compared to other models. Models trained on public datasets are represented in blue, while models trained on private datasets are shown in green. Myna is trained on a publicly-available dataset and is marked in red.

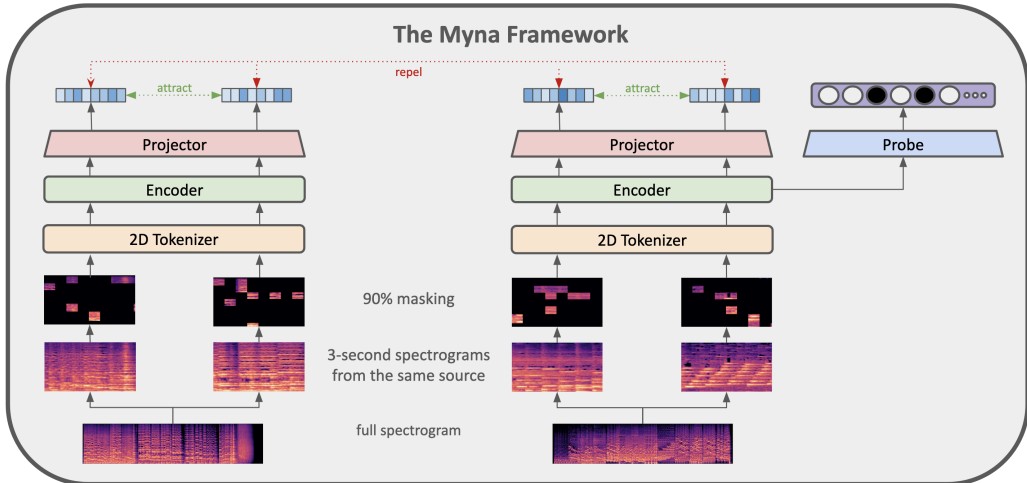

Figure 2: The Myna pre-training framework. Tokens from spectrogram patches are randomly masked before being processed by a transformer encoder. The resulting embeddings are contrasted to maximize similarity between masked views of the same data while minimizing similarity with all other samples (negatives). Tokenizers, encoders and projector modules refer to the same sets of shared weights. For downstream tasks, the projector is discarded and replaced with a task-specific head (labeled "Probe" above) to leverage the learned embeddings.

# 1 INTRODUCTION

The field of Music Information Retrieval (MIR) has been revolutionized by deep learning. Traditionally, tasks such as genre classification, music auto-tagging, chord recognition, and key detection were approached using supervised learning on labeled datasets Pons et al. (2017); Pons & Serra (2019); Choi et al. (2017); Won et al. (2020); Baumann (2021). However, the creation of these datasets is time-consuming and costly, while raw, unlabeled musical data is abundant. This disparity has fueled interest in un- and self-supervised learning, with self-supervised contrastive learning becoming a prominent approach. Recent research has applied frameworks like SimCLR and masked language modeling to extract meaningful musical representations from raw audio or spectrograms Spijkervet & Burgoyne (2021); McCallum et al. (2022); Li et al. (2024); Castellon et al. (2021).

Self-supervised representation learning minimizes reliance on labeled data by learning a rich latent space that can generalize well to downstream tasks. In contrastive learning, the objective is to maximize agreement between different augmented views of the same data while pushing away other pairs of data (negatives). The use of data augmentations is key to contrastive learning; however, traditional data augmentations for musical data do not necessarily give good performance. For example, augmentations such as pitch shifting alter critical musical properties that are essential for tasks like key detection McCallum et al. (2024). Our approach instead relies entirely on token masking to sample different subsets of spectrograms as "views" of the data, which preserves the meaningful relationships between views. We argue that it is more beneficial to teach a model that the relationship between two masked subsets of the input is the same than that two noisy versions of the input are the same; the former keeps the model sensitive to augmentations while the latter makes representations biased to the choice of transformations used. This ensures that we retain musically relevant features while significantly reducing the number of hyperparameters for augmentations (for example, the augmentation chain in CLMR contains 21 hyperparameters Spijkervet & Burgoyne (2021), not including chain ordering; we reduce this to one).

Building on these insights, our work presents Myna [1], a contrastive framework that advances the efficiency of musical representation learning. Myna refines the Contrastive Learning of Musical Representations (CLMR) framework by introducing several key ideas to overcome its limitations.

Our primary contributions are as follows:

---

[1]The name Myna is inspired by the bird native to southern Asia.

- We introduce a simple contrastive learning framework and demonstrate that masking spectrogram tokens can replace traditional data augmentations while maintaining musically relevant features.

- We leverage the ViT architecture to increase memory efficiency and allow for large batch sizes (85x increase in efficiency over CLMR), making training on a single GPU feasible.[2]

- Our model, Myna-Hybrid (22M), trained on a single GPU, achieves competitive results with existing self-supervised approaches, including MULE and MERT-95M-public, highlighting the effectiveness of masking-only contrastive learning.

- When scaled to 85M parameters, Myna-Hybrid further improves across all downstream tasks and surpasses $\text{MuQ}_{m4a}$, establishing itself as the best representation model for global music understanding trained only on publicly available data.

We note that our work is specifically focused on *global* music understanding tasks (tagging, genre, key detection, arousal and valence estimation) rather than lower-level audio processing tasks (such as source separation, transcription, or motif detection), which may require different modeling approaches and objectives.

## 2 RELATED WORK

### 2.1 SELF-SUPERVISED LEARNING FRAMEWORKS

SimCLR Chen et al. (2020) is a simple contrastive approach for learning discriminative representations and has found success in areas ranging from computer vision to language Spijkervet & Burgoyne (2021); Gao et al. (2022). A similar notable framework is Contrastive Predictive Coding van den Oord et al. (2018), a universal approach to contrastive learning, which has been successful for MIR- and audio-related tasks such as speaker and phoneme classification using raw audio. Additionally, this work introduced the InfoNCE loss, which is used in SimCLR, CLMR, and Myna.

Recently, due to the widespread success of transformer-based models on various tasks and modalities, MIR researchers have borrowed unsupervised learning paradigms from natural language processing. In Castellon et al. (2021), the authors probe the hidden layers of OpenAI's Jukebox model Dhariwal et al. (2020) and achieve state-of-the-art results, suggesting that CALM (codified audio language modeling) is an effective pre-training approach for MIR tasks. The authors of this work also suggested that transformer-encoder based models are likely to outperform JukeMIR's performance in music audio representation. Building on this, Li et al. (2024), Won et al. (2023), and Zhu et al. (2025) have emerged as pioneering efforts that harness masked language modeling for musical applications. Masked auto-encoding (MAE) has found success as another non-contrastive pre-training task in images and was recently shown to be effective in environmental sound and genre classification He et al. (2021); Niizumi et al. (2022; 2024).

### 2.2 GENERAL-PURPOSE AUDIO REPRESENTATIONS

The COLA framework Saeed et al. (2020) employs a simple contrastive learning framework built on SimCLR and utilizes Mel-spectrogram representations and bilinear comparisons to achieve better results than supervised counterparts. HARES Wang et al. (2021) further demonstrated that normalizer-free Slowfast networks (trained on the SimCLR objective) lead to effective generalization of audio representations Feichtenhofer et al. (2019); Brock et al. (2021); this finding was later used by McCallum et al. (2022) for music-specific tasks.

### 2.3 PATCH MASKING

While effective in sequence modeling, transformers Vaswani et al. (2017) suffer from quadratic memory and time complexity with respect to the number of tokens. To address this issue, prior work has explored various token masking strategies to reduce computational overhead. In the self-supervised domain, MAE and FLIP He et al. (2021); Li et al. (2023) used masking on image tokens

---

[2]In the contrastive setting, larger batch sizes yield better performance. See Appendix A for batch size ablations.

to increase pre-training efficiency. In the supervised setting, PaSST Koutini et al. (2021) introduced Patchout (spectrogram masking) to speed up transformer training and achieved state-of-the-art results in audio tagging. Our work is the first to show that spectrogram masking works in the contrastive setting.

### 2.4 MUSICAL REPRESENTATIONS

MusiCNN Pons & Serra (2019), a CNN designed for log-mel spectrograms, draws on the discussion in Pons et al. (2017) for its efficient design and is pre-trained on a supervised music auto-tagging task. CLMR Spijkervet & Burgoyne (2021) adapted the SimCLR framework for music using SampleCNN Lee et al. (2018) on raw waveforms and achieved competitive results with supervised counterparts; S3T Zhao et al. (2022) improved on this by using a swin transformer Liu et al. (2021) on spectrograms with simplified augmentations and achieved notable gains in tagging and classification. MULE McCallum et al. (2022) provides a broad analysis of supervised and unsupervised (contrastive) pre-training methodologies on MIR downstream tasks and are the only existing work to not use pitch shifting as an augmentation in a contrastive setting, instead favoring MixUp Zhang et al. (2017) as their sole augmentation. We believe this is a step in the right direction and this work aims to further refine this approach. Their follow-up work studies the effect of various augmentations on model performance McCallum et al. (2024). Recent work has adopted NLP techniques for MIR: JukeMIR Castellon et al. (2021) successfully probed representations from Jukebox Dhariwal et al. (2020), a music generation model based on the GPT architecture. Following this, MERT Li et al. (2024) and MusicFM Won et al. (2023) achieve state-of-the-art results via masked language modeling on music audio tokens.

## 3 METHOD

### 3.1 PRELIMINARIES

Our work builds upon CLMR, which is the music audio adaptation of SimCLR's contrastive learning framework for visual representations. In SimCLR, for every sample $x_i$ in a batch, two augmentations $A(x_i)$ and $A'(x_i)$ are applied, generating two correlated views. These views are passed through the same encoder, and the objective is to maximize agreement between their latent representations using a contrastive loss while minimizing agreement between all other samples in the batch.

SimCLR consists of:

- An encoder $enc(\cdot)$, which maps the augmented views to a latent space $\mathbb{R}^{\text{data}} \mapsto \mathbb{R}^{\text{latent}}$.

- A projector $proj(\cdot)$, mapping latent representations to a projection space $\mathbb{R}^{\text{latent}} \mapsto \mathbb{R}^{\text{proj}}$.

- Stochastic augmentations $A(x)$, producing two correlated views $A(x_i), A'(x_i)$ for each sample.

- A contrastive loss to maximize the similarity between $A(x_i)$ and $A'(x_i)$ and minimize it between views of all other samples.

The contrastive loss used in SimCLR, CLMR, and our work, is the InfoNCE loss van den Oord et al. (2018), defined for a positive pair of examples $(i, j)$ as:

$$\ell_i = -\log\left(\frac{\exp\left(\text{sim}(\mathbf{z}_i^{(1)}, \mathbf{z}_i^{(2)})/\tau\right)}{\sum_{j=1}^{N}\sum_{v=1}^{2} \mathbb{1}_{[j \neq i]} \exp\left(\text{sim}(\mathbf{z}_i^{(1)}, \mathbf{z}_j^{(v)})/\tau\right)}\right)$$

where $\text{sim}(\mathbf{z}_p^{(u)}, \mathbf{z}_q^{(v)})$ denotes the cosine similarity between the normalized representations $\mathbf{z}_p^{(u)}$ and $\mathbf{z}_q^{(v)}$, and $\tau > 0$ is a temperature parameter. Minimizing $\ell_i$ encourages the positive pair $(\mathbf{z}_i^{(1)}, \mathbf{z}_i^{(2)})$ to have a higher similarity than all negative pairs $(\mathbf{z}_i^{(1)}, \mathbf{z}_j^{(v)})$ for $j \neq i$ and $v \in \{1, 2\}$.

---

**Algorithm 1** Myna Pre-Training Algorithm

---

1: **Input:** Unlabeled dataset $\mathcal{D}$, batch size $B$, masking ratio $r$, params $\theta = \{\theta_{\text{enc}}, \theta_{\text{proj}}\}$, lr $\alpha$, temp $\tau$, steps $T$
2: Initialize $\theta$
3: **for** $t = 1$ to $T$ **do**
4:     Sample batch $\{x_i\}_{i=1}^{B} \sim \mathcal{D}$; obtain two segments $s_i^{(1)}, s_i^{(2)}$
5:     Compute Mel-spectrograms $\{m_i^{(j)} = \text{MelSpec}(s_i^{(j)})\}$ and patchify $p_i^{(j)} = \text{Patchify}(m_i^{(j)})$
6:     Mask tokens $v_i^{(j)} = \text{Mask}(p_i^{(j)}, r)$
7:     Encode and project $z_i^{(j)} = \texttt{proj}(\texttt{enc}(v_i^{(j)}; \theta_{\text{enc}}); \theta_{\text{proj}})$
8:     Compute loss $\mathcal{L} = \text{ContrastiveLoss}(\{(z_i^{(1)}, z_i^{(2)})\}_{i=1}^{B}, \tau)$
9:     Update parameters $\theta$ using optimizer (e.g., Adam) on $\nabla_\theta \mathcal{L}$
10: **end for**
11: **return** trained parameters $\theta^*$

---

## 3.2 Creation of Positive Pairs

To generate positive pairs, we first select two three-second segments from the same audio. We generate Mel spectrograms for each segment and then patchify them into $16 \times 16$ or $128 \times 2$ sections. Each spectrogram patch undergoes a linear projection combined with 2D sinusoidal positional encodings Beyer et al. (2022) to create token representations.

Following this, we randomly mask 90% of the tokens from each spectrogram, inspired by the methods in Li et al. (2023) and He et al. (2021). Positive pairs are constructed using the strategy described in Algorithm 1 and illustrated in Figure 2. This masking enables the model to learn meaningful relationships between the remaining tokens, effectively treating the masked spectrograms as augmented views of the same underlying data. The resulting masked pairs serve as positive samples for our contrastive learning framework.

Intuitively, masking a high percentage of tokens encourages the model to focus on global patterns and relationships between the unmasked tokens. By treating masked spectrograms as augmented views, the model is trained to reconstruct meaningful relationships between the unmasked tokens and their masked counterparts. This forces the model to infer higher-level, context-aware features rather than overfitting to specific low-level details that might only be locally relevant. Since the masking process only hides information without altering it (unlike traditional augmentations such as pitch shifting or time stretching), the underlying properties of the music, like pitch/key and BPM, remain intact in both views. This ensures that the model learns representations that are robust to missing information and invariant to the masking operation, allowing it to generalize better to downstream tasks that depend on recognizing the overall structure and relationships in the data.

## 3.3 Why Not Masked Auto-Encoding?

Previous work has demonstrated that masked auto-encoding is an effective pre-training task for learning representations in various domains Niizumi et al. (2022); He et al. (2021). Below, we outline three reasons against using masked auto-encoding for musical representation learning and instead favor a contrastive learning framework.

### 3.3.1 Efficiency

MAE frameworks require training both an encoder and a decoder. While the decoder is necessary for reconstruction during pre-training, it is discarded when transitioning to downstream tasks. This means a substantial portion of computational resources during training is devoted to learning and optimizing a decoder that is ultimately unused. By contrast, our masking-based contrastive learning framework eliminates the need for a decoder entirely and thus reduces computational overhead.

### 3.3.2 TASK DIFFICULTY

In masked auto-encoding, the model is tasked with reconstructing the original input from masked portions, which can be a challenging and sometimes counterproductive objective for music. While MAE has shown success in environmental sound classification, where sounds often exhibit simpler and more repetitive patterns, music exhibits high variability and structural complexity. Musical patterns often span longer temporal contexts, and the relationships between different components (e.g., melody, harmony, rhythm) can be intricate. This makes the reconstruction task disproportionately difficult. Contrastive learning, on the other hand, focuses on learning high-level relationships and invariances rather than predicting low-level details, making it better suited for music (see Appendix B).

### 3.3.3 PRESERVING MUSICALLY RELEVANT FEATURES

MAE forces the model to focus on reconstructing fine-grained details, which may not always align with the musically meaningful features needed for tasks like music tagging, key detection, or emotion recognition. For example, reconstructing the exact values of masked spectrogram tokens could encourage the model to focus on local energy patterns rather than higher-level tonal or rhythmic structures. Contrastive learning emphasizes capturing meaningful global representations, ensuring that the learned features are aligned with the downstream tasks.

### 3.4 MODEL ARCHITECTURE

We use a simplified version of the Vision Transformer (ViT) Dosovitskiy et al. (2020), SimpleViT Beyer et al. (2022), which replaces the CLS token with global average pooling and employs 2D sinusoidal positional encodings. For all experiments in this paper, we use the ViT-S/16 architecture (22M parameters), with the exception of using $16 \times 16$ or $128 \times 2$ non-overlapping patches.

### 3.5 HYBRID MODELS

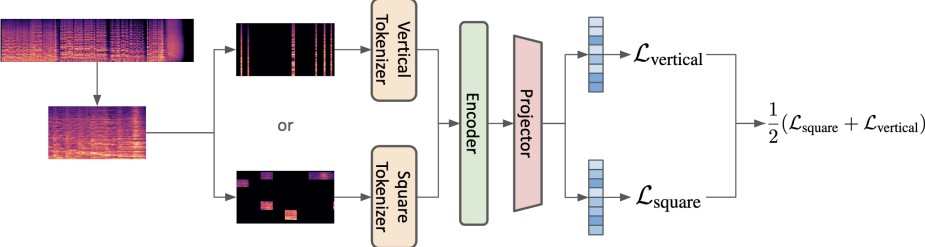

Figure 3: Hybrid model training. A three-second spectrogram is sampled and made into patches. After masking, the patches are processed by their respective tokenizer, consisting of a linear projection and positional embedding. The resulting tokens are fed to a shared encoder/projector module. To compute the hybrid loss, two forward passes are performed with vertical and square patches. The hybrid loss is the average of the vertical and square losses.

Our experiments show that using square ($16 \times 16$) patches yields competitive performance. Conversely, using vertical ($128 \times 2$) patches reduces performance across all metrics, except for key detection, where it achieves state-of-the-art (SOTA) performance among self-supervised methods. To combine the strengths of both approaches, we propose a novel hybrid model compatible with both patch configurations.

The hybrid model retains a shared encoder and projector but employs two separate tokenizers (linear projections) and positional embeddings tailored for the two patch sizes. During training, we alternate between patch configurations. Specifically, at each iteration, we calculate the contrastive loss for each patch configuration independently and then optimize the average of the two losses. The overall objective is $\mathcal{L}_{\text{hybrid}} = \frac{1}{2}(\mathcal{L}_{\text{square}} + \mathcal{L}_{\text{vertical}})$.

where $\mathcal{L}_{\text{square}}$ and $\mathcal{L}_{\text{vertical}}$ are the contrastive losses computed using two separate forward passes with $16 \times 16$ and $128 \times 2$ patches using Algorithm 1, respectively. Figure 3 illustrates the computation process of the hybrid model's loss.

By incorporating this dual-patch training strategy, the hybrid model benefits from the general-purpose performance of square patches while leveraging the superior key detection capabilities of vertical patches. This results in a model capable of excelling across a broader range of musical representation tasks.

## 4 EXPERIMENTS

| Approach | Size | Tags | | Genre | Key | Emotion | | Average |
|---|---|---|---|---|---|---|---|---|
| | | MTT$_{\text{AUC}}$ | MTT$_{\text{AP}}$ | GTZAN | GS | Emo$_{\text{A}}$ | Emo$_{\text{V}}$ | |
| MULE[†‡] | 62M | 91.2 | 40.1 | 75.5 | 64.9 | 73.1 | 60.7 | 68.2 |
| MERT-95M[†‡] | 95M | 91.0 | 39.3 | 78.6 | 63.5 | **76.4** | 60.0 | 68.9 |
| MERT-330M[†‡] | 330M | 91.3 | 40.2 | 79.3 | 65.6 | 74.7 | 61.2 | 69.7 |
| MuQ[†‡] | 310M | 91.4 | 40.1 | **85.6** | 65.0 | 76.1 | **62.8** | **71.4** |
| Jukebox[†] | 5B | **91.5** | **41.4** | 79.7 | 66.7 | 72.1 | 61.7 | 69.9 |
| MusiCNN[*] | 7M | 90.6 | 38.3 | 79.0 | 12.8 | 70.3 | 46.6 | 53.7 |
| CLMR[*] | 3M | 89.4 | 36.1 | 68.6 | 14.9 | 67.8 | 45.8 | 50.8 |
| MERT-95M-public[*] | 95M | 90.7 | 38.4 | 72.8 | 67.3 | 72.5 | 59.7 | 67.7 |
| MuQ$_{m4a}$[*] | 310M | 91.1 | 39.0 | 84.0 | 63.7 | 76.0 | 60.0 | 70.2 |
| MAE[*] | 32M | 88.9 | 35.6 | 75.5 | 53.6 | 69.7 | 50.2 | 62.8 |
| PaSST[*] | 87M | 88.0 | 32.8 | 71.4 | 46.1 | 66.7 | 44.9 | 58.4 |
| Supervised SOTA | N/A | 90.7 | 38.4 | 65.8 | **75.7** | 70.4[◇] | 50.0[◇] | 66.6 |
| Myna-Base[*] | 22M | 90.8 | 39.5 | 78.3 | 63.5 | 73.5 | 55.8 | 67.9 |
| Myna-Vertical[*] | 22M | 90.1 | 37.4 | 75.9 | **68.6** | 66.5 | 45.9 | 66.1 |
| Myna-Hybrid[*] | 22M | 91.0 | 39.8 | 77.9 | **68.0** | 70.8 | 55.2 | 68.6 |
| Myna-85M-Hybrid[*] | 85M | 91.1 | 40.0 | 81.0 | **69.6** | 73.3 | 57.3 | 70.4 |

Table 1: Comparison of Different Approaches on Various MIR Tasks. As in Castellon et al. (2021), tasks with multiple evaluation metrics have their metrics averaged first, and then the averages across all tasks are computed. Models labeled with [*] are trained on publicly-available data, while models labeled with [†] were trained on private datasets. Models marked with [‡] use the MARBLE Yuan et al. (2023) hyperparameter grid instead of the one employed in Castellon et al. (2021) (detailed in Appendix C). All data splits are identical. The max score for all metrics is 100 and higher is better. Note that CLMR was pre-trained on MTT, so its evaluation on MTT does not demonstrate out-of-distribution generalization. Supervised results are from Pons & Serra (2019); Medhat et al. (2017); Baumann (2021); Weninger et al. (2014). ◇ indicates previous supervised works on Emomusic used different dataset subsets for evaluation, and hence numbers are not directly comparable.

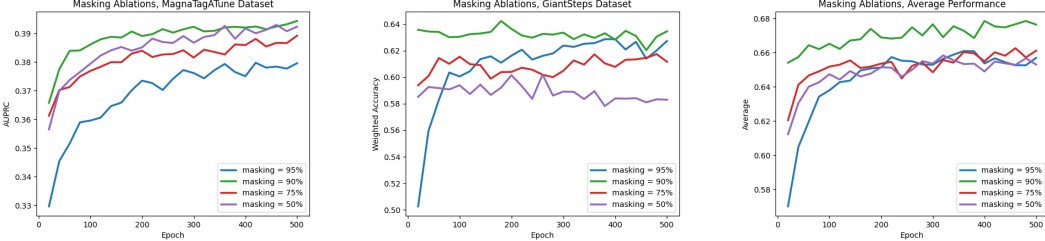

Figure 4: Performance of varying masking ratio on different datasets: MagnaTagATune, GiantSteps, and average across all four benchmarks (MTT, GiantSteps, EmoMusic, and GTZAN).

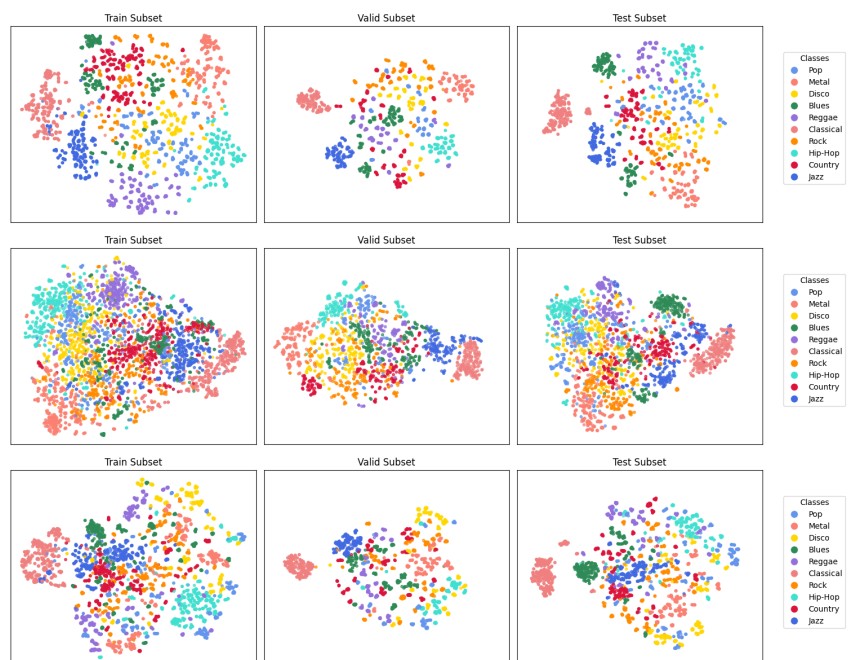

Figure 5: T-SNE visualizations of different embeddings (top to bottom: Myna-Hybrid, MAE, and CLMR) for the GTZAN dataset. Each subplot shows the distribution of samples in the training, validation, and test subsets, with color-coding by class label. The GTZAN dataset was not used in training any of these models.

## 4.1 DATASETS

We pre-train our models on the music subset of the Audioset dataset Gemmeke et al. (2017), containing roughly 822k 10-second music audio segments. Notably, unlike CLMR, we did not train our model on any of the datasets used for downstream tasks (CLMR was pre-trained on the MagnaTagATune dataset). Downstream datasets are detailed in appendix E.

## 4.2 MODELS

We train and evaluate three Myna models with various patch size configurations. Myna-Base, our base model, operates on $16 \times 16$ patches. Myna-Vertical operates on $128 \times 2$ patches. Myna-Hybrid is a hybrid model trained to support both patch sizes simultaneously. During evaluation of Myna-Hybrid, we evaluate a single linear model on the square, vertical, and concatenated (both square and vertical) representations for each task and use the representation that yields the best performance. Hyperparameters are detailed in Appendix F.

## 4.3 RESULTS

For a direct and fair comparison with other approaches, we use the exact data splits, metrics, and evaluation procedure as in Castellon et al. (2021). We briefly summarize the evaluation procedure below for completeness.

To extract relevant information from representations, we employ simple models—linear probes and shallow MLPs—trained on fixed representation vectors to predict task-specific labels. We conduct a grid search over architectures and hyperparameters for each task, varying the model type, hidden dimension, learning rate, and regularization (see Appendix D for more). The model achieving the best performance on a validation set is then evaluated on the test set. This protocol allows for an apples-to-apples comparison of the quality of representations produced by different pre-training strategies.

Based on the results presented in Table 1, Myna demonstrates competitive or superior performance across multiple MIR tasks. Myna-Hybrid (22M) achieves an average score of 68.6, surpassing MERT-95M-public (67.7) and MULE (68.2) while rivaling MERT-95M (68.9). Furthermore, the hybrid model improves performance on music tagging tasks due to its ability to integrate features from both square and vertical patches. Notably, Myna-Vertical and Myna-Hybrid excel in the key detection task with scores of 68.6 and 68.0, surpassing the previous self-supervised SOTA of 67.3. In comparison, Myna-Base and Myna-Vertical exhibit slight trade-offs in performance. Myna-Base delivers robust general-purpose capabilities (67.9 average score), while Myna-Vertical's specialization in key detection (68.6) comes at the cost of lower scores in other areas. Scaling further, Myna-85M-Hybrid achieves the strongest results among publicly trained models, with an average score of 70.4. This not only surpasses MuQm4a (70.2) and MERT-95M-public (67.7), but also rivals much larger private-data models such as MERT-330M and MuQ, highlighting Myna's efficiency and scalability. Interestingly, probing PaSST Koutini et al. (2021) supervised features shows an improvement over MusiCNN (also supervised) in key detection but still has lower performance overall; we hypothesize that the former is due to architecture (Transformer vs. CNN) and the latter is due to domain-specific pretraining. We discuss the effect of masking ratios in Appendix G.

On music tagging and key detection, Myna shows smooth performance improvements with larger batch sizes (see Appendix A for more). However, it underperforms in emotion classification, likely because our masked contrastive learning approach struggles to capture subtle temporal evolution of expressive features, such as gradual shifts in timbre and intensity, which are crucial for emotional perception. Additionally, the Audioset dataset contains 10-second audio snippets that may not contain enough emotional content for the model to recognize. For genre classification, Myna performs well but may be more attuned to harmonic and timbral characteristics rather than the broader rhythmic and structural patterns that often define genres, as contrastive learning emphasizes local spectral similarities over long-term dependencies.

### 4.4 Comparing with Masked Auto-Encoder

MAE, with its focus on reconstructing masked spectrogram tokens, performs well in tasks requiring detailed local information (such as local harmonics that aid in genre classification and many of the tags in MTT, as shown in Table 1) but struggles with tasks that rely on understanding broader musical contexts, such as key detection. Our approach, which instead emphasizes learning global relationships through token masking, consistently achieves stronger generalization across these tasks. For example, in key detection, our model benefits from its ability to capture harmonic relationships without being constrained by the need to reconstruct low-level spectrogram details. This suggests that while MAE excels at learning fine-grained patterns, its objectives may not align with the structural and contextual complexities of music, whereas our approach effectively bridges this gap by focusing on meaningful, high-level representations.

## 5 Future Work

While Myna demonstrates strong performance in musical representation learning, several promising directions remain. First, scaling to larger models and training on more extensive datasets is a natural next step; our initial scaling experiments already suggest that further improvements are likely. Second, we currently sample token subsets for positive pairs uniformly at random. More sophisticated masking policies—either fixed heuristics or learned strategies—may accelerate convergence or yield stronger representations, as has been observed in language and vision pretraining Liang & Larson (2024).

## 6 Conclusion

We introduced Myna, a contrastive learning framework that uses token masking as the sole augmentation strategy. Our approach has shown that this method is effective in learning meaningful representations in the music audio domain while offering significant computational benefits. By leveraging a ViT-based architecture and using token masking as our augmentation, we achieved competitive results with significantly reduced computational requirements. We hope Myna inspires future research to further explore masking-based contrastive learning.

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

## A  BATCH SIZE ABLATIONS

We conduct ablation studies on batch size to investigate its effect on task performance. Results verify previous work Chen et al. (2020) and theory Yuan et al. (2022) that suggests larger batch sizes yield better performance in the contrastive setting. As shown in Table 3, increasing the batch size from 256 to 4096 leads to noticeable and consistent improvements in both individual metrics and the overall average performance. The best results are achieved at the largest batch size of 4096 (Myna-Base), indicating that larger batch sizes are beneficial for achieving optimal performance.

| Approach | Tags | | Genre | Key | Emotion | | Average |
| --- | --- | --- | --- | --- | --- | --- | --- |
| | MTT$_{AUC}$ | MTT$_{AP}$ | GTZAN | GS | Emo$_A$ | Emo$_V$ | |
| Batch size 256 | 90.1 | 38.0 | 75.2 | 60.4 | 68.3 | 52.5 | 65.0 |
| Batch size 512 | 90.3 | 38.3 | 74.5 | 60.7 | 72.4 | 54.5 | 65.7 |
| Batch size 1024 | 90.4 | 38.8 | 74.1 | 61.8 | 69.9 | 56.3 | 65.9 |
| Batch size 2048 | 90.7 | 39.2 | 77.6 | 63.3 | 70.1 | 54.2 | 67.0 |
| Myna-Base (4096) | 90.8 | 39.5 | 78.3 | 63.5 | 73.5 | 55.8 | 67.9 |

Table 2: Performance metrics across various tasks with increasing batch sizes for Myna-Base ($16 \times 16$ patches).

## B  MASKED AUTO-ENCODER VISUALIZATIONS

This section provides visualizations of the Masked Auto-Encoder (MAE) outputs for four randomly selected spectrograms from a held-out validation set. We overlay the output spectrogram with the ground truth unmasked (input) patches to showcase how unmasked patches affect the model's output.

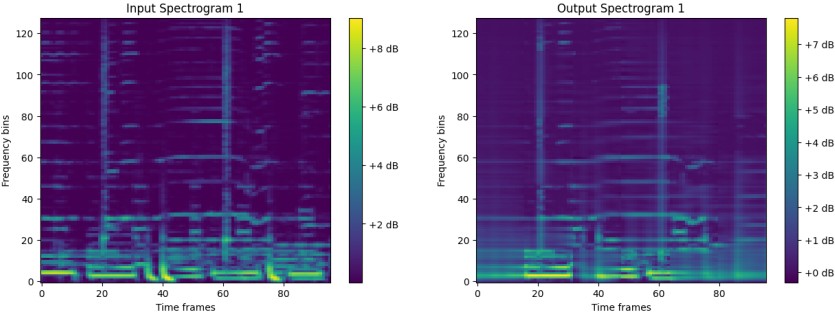

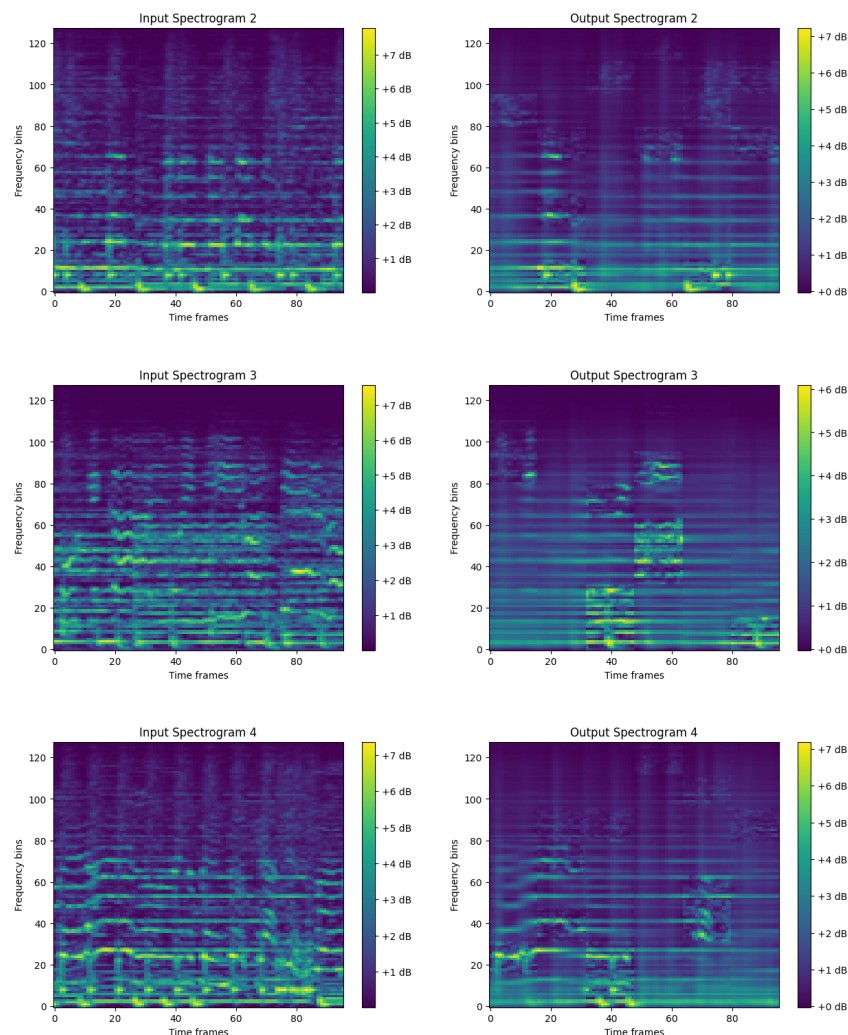

## C MASKED AUTO-ENCODER IMPLEMENTATION DETAILS

In this section, we provide details on the implementation of the Masked Autoencoder (MAE) experiments. We aim for a fair comparison with Myna in terms of architecture, training setup, and evaluation procedures.

### C.1 MODEL ARCHITECTURE

To ensure comparability, we adopted the same encoder architecture as Myna. The encoder model is the same Vision Transformer (ViT-S/16, $16 \times 16$ patches, 22M parameters) identical to the architecture employed by Myna. We keep the same 2D sinusoidal positional encoding Beyer et al. (2022). For the decoder, we used a 6-layer ViT model with the same architecture. To stay consistent with previous work (and because it is the best setting we have found), we used a 75% masking ratio for the input tokens He et al. (2021); Niizumi et al. (2024).

### C.2 TRAINING AND EVALUATION PROCEDURE

We trained the MAE model using the same dataset and hardware setup as Myna, except for differences in training duration (the MAE model took a week to train for the same number of epochs) due to reconstruction overhead. We used a batch size of 1024 as it was the largest batch size that fit on a single A100 GPU. We used the same optimizer (AdamW) and weight decay (1e-5). For a fair

comparison, the MAE model was evaluated identically to Myna. We used the outputs from the last encoder layer as the representation.

# D    EVALUATION PROCEDURE

To evaluate representations for downstream MIR tasks, we follow the procedure as outlined in Castellon et al. (2021): shallow supervised models (linear models and one-layer MLPs) are trained on each task using the representations as input features. A grid search over the following 216 hyperparameter configurations is conducted:

- Feature standardization: {off, on}
- Model type: {Linear, one-layer MLP with 512 hidden units}
- Batch size: {64, 256}
- Learning rate: {1e-5, 1e-4, 1e-3}
- Dropout probability: {0.25, 0.5, 0.75}
- L2 regularization: {0, 1e-4, 1e-3}

Early stopping is applied based on task-specific metrics computed on validation sets, with the optimal model from each grid search evaluated on the task-specific test set. Loss functions are tailored to each task: cross-entropy for genre classification and key detection, independent binary cross-entropy for tagging, and mean squared error for emotion recognition.

# E    DOWNSTREAM DATASETS

**MagnaTagATune (MTT)**: The MTT dataset comprises 25,863 clips, each 29 seconds long, annotated with a set of 188 tags that cover genres, moods, instruments, and other sonic characteristics Law et al. (2009). Similarly to previous work, we use the standard (12:1:3) train, validation, and test split van den Oord et al. (2014); Pons et al. (2017) and do not discard any examples (see Won et al. (2020)). We evaluate using ROC-AUC and average precision (AP) on the top 50 tags.

**GTZAN**: The GTZAN dataset Tzanetakis & Cook (2002), a cornerstone dataset for genre classification in MIR, comprises 1,000 audio tracks, each 30 seconds long, spanning 10 diverse genres. For a fair comparison with previous work, we use the fault-filtered set as described in Kereliuk et al. (2015); Sturm (2013), and report accuracy scores.

**GiantSteps**: The GiantSteps Key dataset Knees et al. (2015) features electronic dance music annotated with key information. It includes roughly 1,000 2-minute song clips covering all 24 major and minor keys (though the data is imbalanced). This dataset challenges models to accurately predict musical keys, which requires sensitivity to harmonic and tonal content. We evaluate using a refined accuracy metric Raffel et al. (2014).

**EmoMusic**: The EmoMusic dataset Soleymani et al. (2013) consists of 744 clips, each 45 seconds long, annotated with valence and arousal scores derived from human listeners' emotional responses. The dataset tests the model's capacity to capture and interpret the emotional cues encoded in music, a sophisticated challenge that probes the depth of the learned musical representations. We report determination coefficients for valence ($R_V^2$) and arousal ($R_A^2$).

# F    HYPERPARAMETERS

We extract Mel spectrograms with 128 bins at a sample rate of 16 kHz using the nnAudio library Cheuk et al. (2020) and apply a 90% masking ratio. The 22M models are trained for 500 epochs (411M examples seen) with a batch size of 4096 on a single NVIDIA A100 GPU, using Adam Kingma & Ba (2017) with a cosine schedule (peak learning rate $3 \times 10^{-4}$, 10 warmup epochs) and weight decay $1 \times 10^{-5}$. Myna-85M-Hybrid is trained with a batch size of 6144 across 4 A100s, a peak learning rate of $1.5 \times 10^{-4}$, and weight decay $2.5 \times 10^{-6}$. For the contrastive loss, we set $\tau = 0.1$. Masking ablations are run on four A100s, as lower masking ratios are less efficient and

require multiple GPUs. While work exists on learning $\tau$ via gradient descent Radford et al. (2021) or individualized temperature values Zi-Hao Qiu & Yang (2023), we keep it constant in this work.

# G MASKING RATIOS

We investigate the impact of varying the masking ratio on model performance. As shown in Figure 4, we find that increasing the masking percentage generally improves performance. However, performance saturates at 90%, as pushing it to 95% removes too much information and leads to performance degradation. Additionally, we note a clear correlation between the masking ratio and the model's average performance, and suspect that low masking ratios make the contrastive task too easy, which leads to less discriminative (and thus useful) representations. This is particularly advantageous since increasing the masking ratio also improves computational efficiency by reducing the number of tokens that the model needs to attend to.

We qualitatively evaluate Myna's discriminative capacity against MAE and CLMR on the GTZAN dataset in Figure 5. Myna demonstrates clearer separation between classes with noticeably reduced overlap between class clusters; this indicates that Myna's embeddings capture more meaningful and discriminative features.

