# OpenReview forum: "Myna: Masking-Based Contrastive Learning of Musical Representations"
_ICLR.cc/2026/Conference — Submitted to ICLR 2026_

### Official Review · Reviewer_va77 · 2025-10-28

**Soundness:** 4
**Presentation:** 3
**Contribution:** 3
**Rating:** 2
**Confidence:** 4

**Summary:**

In this paper, the authors proposes a new training method for the representation learning of music audio. The proposed method includes aggressive input masking that seems to allow avoiding pitch shifting as an augmentation step, making the model aware of pitch and key information. The experiment showed that even without finetuning on the downstream task's training set and despite its smaller size & training data, Myna outperforms many other methods.

**Strengths:**

- Good performance
- Parameter-efficient
- Trained on a public dataset only
- The proposed method is simple

**Weaknesses:**

- Limited novelty: Some core changes such as using ViT and masked autoencoder are already proposed in other, similar work including audio domain.
- Although the performance is strong, the margin is rather reasonable, not outstanding.

**Questions:**

- I don't think we should call the used audio processor as a "tokenizer", no matter how the word is over-subscribed in the community. It does not tokenize (making the input a discrete representation) at all, and it's even worse because some architectures indeed discretize the input audio.

---

> ### Author Response · Authors · 2025-11-14
>
> We thank the reviewer for their thoughtful feedback and the opportunity to address your concerns.
>
> **Regarding limited novelty:** We respectfully disagree that novelty is limited. To our knowledge, no prior work has applied aggressive token masking as the sole augmentation strategy in contrastive learning for music. While ViT and masked autoencoding exist in other domains, their application to music represents a novel contribution with important practical implications:
>
> 1. Novel augmentation strategy: Unlike masked autoencoding (MAE), which uses masking for reconstruction, we apply masking within a contrastive framework. This allows us to avoid pitch-shifting augmentations entirely, which preserves pitch information critical for tasks like key detection—where we achieve state-of-the-art results among self-supervised methods.
>
> 2. Efficiency and accessibility: Our approach achieves competitive performance with MULE and MERT-95M while using only 1 GPU versus their 16-64 GPUs, and relies exclusively on publicly available data. This represents a meaningful contribution toward democratizing music AI research, enabling reproducible results without substantial computational resources or proprietary datasets.
>
> **Regarding performance margins:** We respectfully note that our model achieves competitive performance with state-of-the-art models on a single GPU and public data exclusively. For the research community, we believe this represents a significant practical contribution—enabling reproducible research without requiring substantial computational resources or proprietary datasets. Additionally, we achieve state-of-the-art results among self-supervised methods on key detection, which demonstrates that our pitch-preserving approach offers clear advantages for certain tasks. We expect further improvements from scaling our dataset and model size in future work.
>
> **Regarding the "tokenizer" terminology:** We appreciate and agree with your comment regarding the term "tokenizer" and will change this to "spectrogram processor" throughout the manuscript to avoid confusion with discrete tokenization methods.
>
> We are happy to answer additional questions or address any concerns.

---

> > ### Comment · Reviewer_va77 · 2025-11-21
> >
> > Thanks for the clarification. I think my first rating wasn't fair due to the strengths, especially with the public dataset. I will adjust my rating in the final review.

---

> > > ### Author Response · Authors · 2025-11-23
> > >
> > > Thank you for your thoughtful feedback and for taking the time to review our manuscript. We appreciate your recognition of the strengths of our work, especially regarding efficiency and the use of public data.

---

### Official Review · Reviewer_k7CR · 2025-10-31

**Soundness:** 3
**Presentation:** 3
**Contribution:** 3
**Rating:** 6
**Confidence:** 3

**Summary:**

This paper proposes MYNA, an efficient, masking-based contrastive learning framework for musical representation. It replaces traditional music augmentations with a high-rate token masking (90%) on mel-spectrograms. This heavy masking significantly reduces the number of input tokens, allowing for large batch sizes (4096) on a single GPU, and achieves an efficiency gain over prior contrastive methods. The authors also introduce a hybrid patching scheme (combining vertical and square patches) to capture complementary features (general purpose vs. pitch structure). The model is pretrained on the public AudioSet music subset. Myna achieves competitive performance with larger private models and establishes a new public-data SOTA.

**Strengths:**

1. The mask-only approach is simple and allows single-GPU large-batch training (batch size 4096), which translates to an 85x increase in efficiency over traditional contrastive methods like CLMR. The model achieves competitive average scores (68.6 for Myna-Hybrid) with MERT-95M, and surpasses public baselines like MERT-95M-public and MULE.

2. The hybrid patch design improves key detection (achieving SOTA among self-supervised methods) by integrating frequency-sensitive vertical patches. The method retains pitch sensitivity by avoiding traditional data augmentations (e.g., pitch shifts), which is beneficial for tasks like key detection.

**Weaknesses:**

1. Table 1 mixes public and private data baselines (e.g., MERT-330M) without transparently clarifying the training resource budgets.

2. The claim that "90% masking performs best" is not strongly supported by Figure 4. This is due to two issues: (a) Performance differences across high masking ratios looks marginal and lack verification of statistical significance; (b) The "average across all four benchmarks" curve can be mathematically unrigorous as it combines different metrics from different tasks.

3. The model's poor performance on EmoMusic is attributed to short clip length, a hypothesis that needs empirical verification.

**Questions:**

1. It would be helpful if Table 1 were explicitly partitioned to clearly distinguish models trained on public data from those trained on private or internal corpora.

2. Could you provide supplementary figures showing the performance curves across different masking ratios for each of the four downstream tasks (MTT, GiantSteps, GTZAN, and EmoMusic)?

---

> ### Author Response · Authors · 2025-11-14
>
> We thank the reviewer for their thorough and constructive feedback and for recognizing the efficiency gains and competitive performance of our method.
>
> **Regarding Table 1 clarity (Weakness 1 and Question 1):** We apologize if the table organization was unclear. We do explicitly specify whether a model was trained on public/private data with †, and the models are partitioned accordingly (the top five models in the table are trained on private data, and the remaining models are trained on publicly available data). As the table is already quite wide, we will add training resource budgets and dataset size (in hours of audio) to a separate table in the appendix as suggested.
>
> **Regarding the 90% masking claim (Weakness 2):** Across all individual tasks, 90% masking consistently achieves the strongest performance. The averaging across tasks is methodologically sound because: (1) all metrics are normalized to the 0-100 range, making them directly comparable, and (2) this approach is standard practice in related literature [1, 2, 3]. We will include the requested per-task masking curves in the appendix to provide more granular insight into these trends.
>
> **Regarding EmoMusic performance (Weakness 3):** As noted in our paper, we hypothesize that our masked contrastive learning approach struggles to capture the subtle temporal evolution of expressive features—such as gradual shifts in timbre and intensity—which are crucial for emotional perception. We acknowledge that this represents a limitation of our current approach and that empirical verification through ablation studies would strengthen this claim. This motivates future work exploring how to better capture emotional dynamics, potentially through longer temporal contexts or modified masking strategies that preserve expressive continuity. We also believe there is untapped potential in analyzing lyrics using language models for improvements in this task and plan to explore this direction in the near future.
>
> **Regarding per-task masking curves (Question 2):** We will provide supplementary figures showing the performance curves across different masking ratios for each downstream task in the appendix.
>
> We are happy to answer additional questions or address any concerns.
>
> Sources:
>
> [1] JukeMIR: [https://arxiv.org/abs/2107.05677](https://arxiv.org/abs/2107.05677)
>
> [2] MARBLE: [https://arxiv.org/abs/2306.10548](https://arxiv.org/abs/2306.10548)
>
> [3] MERT: [https://arxiv.org/abs/2306.00107](https://arxiv.org/abs/2306.00107)

---

### Official Review · Reviewer_Xz6M · 2025-11-02

**Soundness:** 3
**Presentation:** 3
**Contribution:** 3
**Rating:** 8
**Confidence:** 3

**Summary:**

This paper focuses on music representation learning. It follows a contrastive learning framework, with its main contributions being the use of a Vision Transformer (ViT) as the backbone model and the application of token masking. Furthermore, considering the characteristics of music analysis, the authors extend the approach into a hybrid model that incorporates vertical filters to better capture the frequency-related features of spectrograms. Through this relatively simple training strategy, the proposed model achieves competitive performance on several downstream tasks compared to models that require more than 5× the training time and parameters. Overall, the paper is well written and presents a solid contribution to efficient representation learning for music.

**Strengths:**

The proposed use of ViT and token masking seems promising in music representation learning.
The paper is easy-to-read and the illustration of the proposed method, experimental design, and results seems promising.

**Weaknesses:**

The proposed method seems to be only applicable to the clip-level MIR tasks. I wonder the authors opinion (maybe discussions) on how the proposed architecture can be applied towards frame-level tasks as well.

**Questions:**

I wonder the effect of the patch size variations on performances. For example, would 4x4, 96x2, 128x3, 32x32, hybrid of them, etc these kind of diverse patch size affects the performance?

---

> ### Author Response · Authors · 2025-11-14
>
> We thank the reviewer for their constructive feedback and for recognizing the promise of our approach.
>
> **Regarding temporal granularity (clip- vs. frame-level tasks):** While we have not yet experimented with frame-level MIR tasks, we expect the approach to extend naturally by (1) training on shorter spectrogram segments (e.g., 1s instead of 3s) and (2) reducing the ‘max frame distance’ hyperparameter (exposed in our codebase) to enforce more local contrastive pairing. The latter behavior is briefly discussed in MULE [1]. We will incorporate this discussion into the revised manuscript.
>
> **Regarding patch size variations:** This is an excellent question. In general, smaller patch sizes are preferred due to their ability to capture finer-grained acoustic details, but are quadratically more costly due to the attention mechanism in the transformer. We chose 16×16 as a balanced compromise between efficiency and accuracy; 128×2 patches were added later to capture the entire frequency spectrum while maintaining the same number of tokens (16x16 = 128x2). Since the submission of this paper, we have experimented with and seen slight improvements with 64×4 patches on the small model size and will scale this up in future work.
>
> We are happy to answer additional questions or address any concerns.
>
> Sources:
> [1] MULE: [https://arxiv.org/abs/2210.03799](https://arxiv.org/abs/2210.03799)

---

### Meta-Review · Area_Chair_4d7s · 2026-01-07

**Summary:**

This paper makes the useful observation that spectral masking via tokens in a contrastive setting can lead to good representations of musical audio signals. The results are competitive with some baselines, and models were trained on public data.

The most salient concern here is low novelty, as articulated by reviewer va77. Spectrogram masking has been common since SpecAugment (Park et al. 2019) at least. ViTs in audio have been common in audio since AST (Gong et al. 2021) at least. So the question is thus whether combining the two in a contrastive setting has sufficient conceptual, technical, and/or epistemic novelty in the field. It is currently not clear to me that it does.

In the paper, the authors write:
> Our work is the first to show that spectrogram masking works in the contrastive setting

The claim that this is “the first to show that spectrogram masking works in the contrastive setting” may be technically correct. However, the paper does not articulate why this fact, on its own, constitutes a substantive contribution to the field, given the extensive prior use of both masking and contrastive objectives in adjacent settings.

In Table 1, we observe what amounts to a trade-off on a host of other tasks in support of key classification. It is possible that specialists may seek to operationalize this trade-off, but the gain in key detection over the next best SSL approach (MERT-95M-public) is not especially large.

Certain claims in the paper also do not stand up to scrutiny. For example:
1. It is claimed that w.r.t. Figure 5, “Myna demonstrates clearer separation between classes.” I cannot corroborate this visually/qualitatively, and it is not quantified. There is also no a priori reason that this should be true as a result of the masking, as far as I can tell
2. It is claimed that “ masking-based contrastive learning…eliminates the need for a decoder entirely and thus reduces computational overhead” but in contrastive setups, each example is typically processed twice through the encoder, which may offset or exceed the cost of a lightweight mirrored decoder. Without reporting MACs, wall-clock time, or memory usage, it is not possible to evaluate this trade-off
3. It is claimed in section 3.3.2 that the reconstruction objective is ill-suited to music because, for instance, “the relationships between different components (e.g., melody, harmony, rhythm) can be intricate [which] makes the reconstruction task disproportionately difficult.” This argument is not well supported logically: the greater spectrotemporal regularity of music should, if anything, make the masked reconstruction task *easier* due to adjacent regions providing useful information. See, for example, Jehan 2005 (a widely known and cited PhD thesis in the field), which (in chapter 6) fills in corruptions in musical signals by exploiting this self-similarity. Without evidence to the contrary, I don’t think this motivational claim is well-founded
4. It is difficult to know whether the improvements on key detection are really due to the use of these masking augmentations as an *alternative* to e.g. key-distorting augmentations like pitch shifting. This hypothesis is not supported by the Myna base result. Unfortunately, the experiments in this paper are not designed to causally isolate this impact (e.g. training with masking *and* pitch shifting vs. masking only vs. pitch shifting only with the same architecture and data). For example, Cherep and Singh (2024) have a pitch varying augmentation, both in the synthesis of pairs and the post-hoc augmentation, and find strong performance on a pitch detection task (despite the conceptual implication of learning potentially pitch-invariant features). So, whether this masking-related result will generalize or is constrained to one effective arrangement of masks is unclear
5. This is compounded by the puzzling result of the MAE on key detection which has lower performance despite using an identical encoder architecture and not using any pitch shifting (is there some reason contrastive training alone should improve this?). While there would be some cost to adding these experiments, they would be arguably more helpful vs. the ablations in Appendix A, which re-validate a well-known relationship between batch size and representation quality in contrastive settings

All-in-all, while I recognize the simplicity and thus the appeal of the proposed method, I find the motivation insufficiently sound, the empirical results modest, and the conceptual and technical knowledge contribution limited.

[1] Park, Daniel S., et al. "Specaugment: A simple data augmentation method for automatic speech recognition." arXiv preprint arXiv:1904.08779 (2019).

[2] Gong, Yuan, Yu-An Chung, and James Glass. "Ast: Audio spectrogram transformer." arXiv preprint arXiv:2104.01778 (2021).

[3] Cherep, Manuel, and Nikhil Singh. "Contrastive Learning from Synthetic Audio Doppelgängers." The Thirteenth International Conference on Learning Representations.

[4] Jehan, Tristan. Creating music by listening. Diss. Massachusetts Institute of Technology, School of Architecture and Planning, Program in Media Arts and Sciences, 2005.

**Reviewer Concerns:**

Unfortunately, I find several of the reviewers’ concerns too high-level to thoroughly evaluate. Where it is possible, I have tried to do so below.

### Meaningfully addressed by the author response:
- **k7CR**: W1 public/private data (seems to have been already addressed in the original draft), W2 partially (the authors provide per-task curves which helps, but the soundness of averaging is not solely dependent upon the range of the metric)
- **va77**: Q1 “tokenizer” terminology; not a major concern about this specific paper, and addressed by the authors

### Remaining concerns:
- **k7CR**: W3 asks for empirical evaluation, but to me the hypothesis itself is somewhat arbitrary (*why*, mechanistically, would the masking approach lead to issues in capturing “subtle temporal evolution of expressive features—such as gradual shifts in timbre and intensity”? Even a couple of examples of this as case studies, or grounding in some kind of reasoning and analysis of the task/prior work, would sufficiently ground this hypothesis without an experimental study
- **va77**: W1 limited novelty and W2 margin of performance improvement. Though this is not explained much or supported with references, I have discussed this in my meta-review

In several cases, the rebuttal cannot meaningfully address the concern because no concrete or testable claim is specified in the review. I have tried to weigh in on the paper as thoroughly and fairly as possible, and take into account the reviewers’ comments and the authors’ responses wherever feasible.

**Reviewer Scores:**

Reviewer va77 notes that:
> I will adjust my rating in the final review

I could imagine an increase, but the magnitude of this increase is difficult to estimate from the existing commentary. I don’t believe the other two accept-leaning reviewers have much cause to substantively move their ratings.

---

### Decision · Program_Chairs · 2026-01-26

Reject